# A simple method for the calculation of dialysis Kt factor as a quantitative measure of removal efficiency of uremic retention solutes: Applicability to high-dialysate vs low-dialysate volume technologies

**Giacomo Colussi**[1,2☯]*, **Chiara Carla Maria Brunati**[1☯], **Francesca Gervasi**[1,3], **Alberto Montoli**[1], **Denise Vergani**[1], **Federica Curci**[1], **Enrico Minetti**[1]

**1** Division of Nephrology, Dialysis and Renal Transplantation, ASST GOM Niguarda, Milan, Italy, **2** Ambrosiana Clinic-Sacra Famiglia Foundation, Milan, Italy, **3** Nephrology School, Milano-Bicocca University, Milan, Italy

☯ These authors contributed equally to this work.
* giacomo.colussi@ospedaleniguarda.it, giacomo.colussi@fastwebnet.it

## Abstract

Dialysis urea removal metrics may not translate into proportional removal efficiency of non-urea solutes. We show that the Kt factor (plasma volume totally cleared of any solutes) differentiates removal efficiency of non-urea solutes in different technologies, and can easily be calculated by instant blood-dialysate collections. We performed mass balances of urea, creatinine, phosphorus and beta$_2$-microglobulin by whole dialysate collection in 4 low-flux and 3 high-flux hemodialysis, 2 high-volume post-hemodiafiltration and 7 short-daily dialysis with the NxStage-One system. Instant dialysate/blood determinations were also performed at different times, and Kt was calculated as the product of the D/P ratio by volume of delivered dialysate plus UF. There were significant differences in single session and weekly Kt (whole dialysate and instant calculations) between methodologies, most notably for creatinine, phosphorus and beta$_2$-microglobulin. Urea Kt measured in balance studies was almost equal to that derived from the usual plasma kinetic model-based Daugirdas' equation (eKt/V) and independent V calculation, indicating full correspondence. Non-urea solute Kt as a fraction of urea Kt (i.e. fractional removal relative to urea) showed significant differences between technologies, indicating non-proportional removal of non-urea solutes and urea. Instant Kt was higher than that in full balances, accounting for concentration disequilibrium between arterial and systemic blood, but measured and calculated quantitative solute removal were equal, as were qualitative Kt comparisons between technologies. Thus, we show that urea metrics may not reliably express removal efficiency of non-urea solutes, as indicated by Kt. Kt can easily be measured without whole dialysate collection, allowing to expand the metrics of dialytic efficiency to almost any non-urea solute removed by dialysis.

**Data Availability Statement:** All relevant data are shown in the article and its Supporting Information files.

**Funding:** The authors received no specific funding for this work

**Competing interests:** I have read the journal's policy and the authors of this manuscript have the following competing interests: Drs.GC and CCMB have been past medical advisors for NxStage Medical Inc, Lawrence, MA; Drs.CCMB and EEM received fees as speaker in dialysis industry-sponsored symposia at the National Congress of Italian Society of Nephrology. Drs. AM, FG, DV and FC have no conflict of interests to declare. This does not alter our adherence to PLOS ONE policies on sharing data and materials.

## Introduction

Mass of urea removed by any dialytic modality is the current standard for measuring dialysis efficiency and dose. Individual indices in use may differ, e.g. spKt/V, eKt/V or URR in single sessions, or on weekly basis stKt/V, solute removal index (SRI) or (for PD) weekly Kt/V [1]. Given information is equivalent for all of these indices, and consists in the volume-equivalent of mass of urea removed as a fraction of body urea pool taken as either an averaged or a maximal/peak value [1]. Availability of a solid yet simple mathematical modelling allowing their calculation by only a couple of blood urea concentration measurements and a few anthropometric patient data, specifically avoiding direct dialysate measurements, has greatly contributed to almost universal spread and adoption of Kt/V indices as a standard in dialysis metrics. Weekly extrapolation as stKt/Vis also a standard for comparisons between technologies with different frequencies and/or total time on treatment (e.g. DP vs HD, or thrice-weekly vs "more frequent" HD).

Use of urea-based indices is not without inconveniences. Kt/V represents more a patient-specific index of "adequacy" or "dose" than of technology "efficiency", since factorization is made by patient factors, which may greatly differ between patients. Just to exemplify, removal of the same quantity of urea would result in different Kt/V in 2 patients with similar plasma levels but different weights. Thus, for a more objective comparison between different dialysis modalities and technologies the factor "Kt" (i.e. volume of blood totally cleared of any individual solute) might be more appropriate than the "Kt/V" [2]. Operationally, this might simply be done by extrapolating "V" from anthropometric information and correcting corresponding Kt/V-based indices.

Even more important, it is well acknowledged that urea is only a surrogate for the uremic toxin family, and that removal efficiency of non-urea solutes in individual technologies may not be proportional to that of urea. One may anticipate that membrane characteristics may affect differently the removal of low molecular weight (as urea) and higher molecular weight solutes, and that time of treatment and frequency may also have differential effects on solutes with different distribution volumes, such as urea as compared to phosphate and beta$_2$-microglobulin (b$_2$M). Unfortunately, there are no reliable models to evaluate removal efficiency from dialysis-associated changes in plasma levels for solutes other than urea.

We have recently being comparing removal efficiency of different solutes between the novel NxStage System One (NSO) technology, based on frequent (6 times/week), short treatments and low volume dialysate, with traditional hemodialysis technologies, i.e. high-flux bicarbonate dialysis (BHD), post-dilution high volume hemodiafiltration (HDF) and automated nocturnal peritoneal dialysis (APD) [3, 4]. Mass balance of each solute was measured by whole dialysate collection in all technologies; the plasma volume-equivalent of removed mass of each solute (i.e. the Kt factor) was easily calculated either by dividing the total quantity removed by time-averaged plasma concentration (TAC), or as the product of dialysate volume multiplied by the mean dialysate to plasma (D/P) concentration ratio (i.e. the ratio between mean dialysate concentration and TAC). We were able to verify that in each extracorporeal treatment modality the D/P ratio of each solute remained stable throughout the treatment, and very close to the mean dialysate concentration to TAC ratio; thus we reasoned that any instant D/P ratio along a session might allow to measure the Kt factor of any solute *X* simply by multiplying that ratio by total dialysate volume. The latter is reliably indicated by monitors as the sum of delivered dialysate and total ultrafiltration. Thus the mass (as volume-equivalent) of any solute cleared off by dialysis may be directly measured by a single, or better a few, instant blood and dialysate collections; the practical issue is that additional solutes may complement urea as indices of dialysis efficiency, allowing to enlarge comparison metrics between technologies.

In the present investigation, we have compared the Kt factor of different solutes, namely urea, creatinine, phosphate and $b_2M$, in different dialytic modalities: bicarbonate dialysis with low-flux or high-flux membranes ($BHD_{lf}$ and $BHD_{hf}$), high-volume post-dilutions hemodiafiltration (HDF) and short-daily hemodialysis with the NxStage system (NSO). We show that individual solute removal efficiency, as indicated by Kt, differs between technologies; additionally, removal efficiency of non-urea solutes does not show equal proportionality to that of urea in different technologies. Lastly, we show that instant D/P measurements along treatment in long-lasting, high volume-dialysate technologies allow quantitative comparisons of removal efficiency between technologies without the need of all-treatment dialysate collection, and also allow to reliably calculate total quantitative solute removal.

## Materials and methods

In 4 $BHD_{lf}$, 3 $BHD_{hf}$ and 2 HDF we performed full balance studies with collection of all-treatment dialysate as described [3, 4]; in short, all effluent dialysate was collected in 6 plastic bags, each one for every 40 min treatment-time, weighted with precision electronic balances and well mixed before sample collections. Instant dialysate and blood samples were also collected for instant D/P ratio at 60, 100, 140 and 180 min from dialysis start. Later on the timing for instant collections was standardized at 60, 120 and 180 min. Start and end-treatment (after 5 min low-blood flow and stop-dialysate) blood samples were also performed for TAC calculations (see below). In each blood and dialysate samples (bag and instant) urea, creatinine, phosphorus and $b_2M$ were measured. Seven NSO treatments were also evaluated with all-treatment dialysate collection and a single instant dialysate and blood collection at 60; in a single patient instant evaluations were also performed at 30 and 120 min.

All treatments were performed according to "standard-of care" prescription in our Centre: BHD and HDF lasted 4 hours/run, thrice a week, with a constant ultrafiltration along the run according to programmed total weight loss, QB of at least 300 ml/min, QD 500 ml/min (BHD) or 600 ml/min (HDF), of which a quote of about 100 ml/min was used for reinfusion; "equilibrated" single session Kt/V (eKt/V) was targeted to 1.2 or more.

NSO treatments were programmed for 6 days/week with a 40% ratio of dialysate to blood flow, and single session dialysate delivery calculated as 0.5*TBW/0.85 liters, where TBW is total body water according to Watson's formula and 0.5 represents single session spKt/V [3, 4], predicted weekly stKt/V being 2 or more. Dialysate was generated on-line in BHD and HDF and was delivered as premixed sterile non pyrogenic bags in NSO; final composition was (in mM): Na 140, K 2, Ca 1.5, Mg 0.5, acetate 3, bicarbonate 34 (BHD) and 37 (HDF); in NSO the base was Lactate (45 mM) without any acetate. Membranes in use were Polyamix$^{TM}$ low-flux in $BHD_{lf}$ (Polyflux Baxter/Gambro L series, nominal Kuf 15 ml/hour*$m^2$*mmHg and surface area 2.1 mq), Polyamix$^{TM}$ high-flux in $BHD_{hf}$ (Polyflux Baxter/Gambro H series, Kuf 70, 2.1 mq) and polyhetersulfone high flux in NSO (Purema$^{TM}$ H series, Kuf 85, 1.6 mq).

## Calculations

All plasma solute concentration values are given as plasma water levels according to Colton's formula [5]: $PXpw = (1–0.0107 * TP)$, where $P_X$ is plasma concentration of solute $X$ and TP is total plasma protein concentration in g/dl. End-treatment values were also corrected for hemoconcentration according to Bergstrom an Wehle [6]:

$$PXcor = PXpwpost / \left( 1 + \frac{UF}{0.2 * BWpost} \right)$$

where UF is dialytic weight loss, BW is body weight and suffix "post" indicates the end of

dialysis. This value was further corrected for solute compartment disequilibrium (post-dialytic rebound) according to Tattersall [7]:

$$PXeq = PXpwpre * \left(\frac{PXcor}{PXpwpre}\right) \wedge \left(\frac{t}{t + Tx}\right)$$

where $t$ is treatment time and tx a specific equilibration time (min) for each solute: 35 for urea, 50 for creatinine, 60 for phosphorus and 110 for $b_2M$. $P_{Xpw}eq$ was used for calculations of TAC; mean study D/P ($D/P_{bal}$) was derived from estimated solute concentration in all spent dialysate (i.e. measured total quantity over measured total volume) and TAC.

TAC along full studies ($TAC_{bal}$) for each solute X was calculated as [8]:

$$TACXbal = (PXpwpre - PXeq)/Ln\left(\frac{PXpre}{PXeq}\right)$$

where start (pre) and end-treatment are all plasma water values, with correction for hemoconcentration and equilibration, as already stated.

Plasma water values were used also for instant D/P ($D/P_{inst}$), but TAC from instant data ($TAC_{inst}$) was calculated according the above formula with plasma water values at 60 and 180 min, without any correction.

Kt (liters/session) in full balances ($Kt_{bal}$) was calculated as the ratio of measured quantity of each solute X in spent dialysate ($Q_{bal}$) and corresponding $TAC_{bal}$. In instant calculations, Kt ($Kt_{inst}$) was instead the product of measured instant D/P (either as individual value at 60 min in NSO or the mean of the 3–4 serial determinations in BHD/HDF) by dialysate volume (liters), indicated by monitors as delivered dialysate plus UF (BHD and HDF), or direct measure (NSO). In instant calculations quantity of removed solutes ($Q_{inst}$) was calculated as the product of $Kt_{inst}$ by $TAC_{inst}$ in BHD/HDF and by the 60min blood concentration ($P_{60}$) in NSO.

To check correspondence of our Kt metrics with the plasma kinetic model in current use, urea Kt was also extrapolated from the blood-based Daugirdas second generation eKt/V equation [9]. Since Daugirdas equations do not allow to separately solve for K and V, but only for the K/V ratio in a given time t, V was separately calculated on the basis of total urea removed in spent dialysate ($Q_{bal}$), start and end urea levels, and total UF along treatment according to formula:

$$V(liters) = \frac{Qbal}{Pureapre - Pureaeqpost} - Pureapre * \frac{UF}{Pureapre - Pureaeqpost}$$

(see Appendix for details). Since plasma urea entries in Daugirdas equations are not corrected for plasma water, uncorrected values were also used for V calculation.

Treatments included in present analysis were performed from November 2018 to June 2019 in clinically stable adult patients, on renal replacement therapy for at least 6 months, undergoing their standard prescribed therapy. Blood access was by native fistulas in all, with no consistent recirculation.

In our institution all dialysis patients routinely sign a written informed consent for the specific dialysis treatments chosen as replacement therapy, for collection of blood samples necessary for monitoring treatment adequacy and efficacy in time, and periodically shared with patients for any therapy updates, for collection of all data in personal clinical charts and for their anonymous use in aggregated analysis and any research studies. Data of interest were collected along August to September 2019 from patient electronic and paper records preserved at the Dialysis Unit of the Niguarda Hospital and transferred as encoded (treatment type and

number), anonymous files into an Excel spreadsheet before analysis. As a retrospective, observational and anonymous study, no approval by local Ethic Committee was required. This investigation complies with the principles of the Helsinki declaration.

All biochemical determinations were performed by routine methods in our central laboratory, which operates within the CISQ Network (Italian federation of Management System Certification Bodies, Certificate n. 9122.AONI).

## Statistics

Data are shown as numbers and mean±SD; statistical differences between matrices of D/P ratios and Kt at different times was checked by ANOVA, followed by Scheffé post-hoc test if a statistically significant difference was shown. Coefficient of variation (CV) of each set of instant D/P and Kt within a single session was calculated for individual patients, and for all the studies as the mean±SD of individual CVs. Instant data in each patient were compared to corresponding whole-balance data by paired Student's t test. Correlation between instant data and whole-balance results in each patient was performed by linear regression analysis by use of Pearson's r coefficient. A two-tailed $p < 0.05$ was considered statistically significant. No sample size and power calculations were performed for this pilot study, due to lack of sound criteria for defining meaningful differences between calculations. All evaluations were performed by use of a Microsoft Excel 2010 spreadsheet.

## Results

$D/P_{inst}$ for all solutes remained substantially stable along the run, without significant differences at any time (Fig 1); intra-session CVs were 8.6±3.6% (urea), 16.5±7.3% (creatinine), 13.5 ±8.5% (phosphorus) and 10.6±6.25 ($b_2M$).

Mean $D/P_{inst}$ of all solutes was significantly higher than corresponding $D/P_{bal}$ both in aggregated BHD/HDF (Table 1) and NSO (Table 2). Also $Kt_{inst}$ was substantially stable along sessions, and was significantly higher for any solute than corresponding $Kt_{bal}$ (Tables 1 and 2)

There was a highly significant correlation between $Kt_{inst}$ and $Kt_{bal}$ for all evaluated solutes, with correlation lines almost parallel to the identity lines, suggesting a systematic, methodology-related cause for this difference (Fig 2). $Q_{inst}$ almost matched measured $Q_{bal}$, with no or very small differences (Tables 1 and 2); correlation equation of individual $Q_{inst}$ and $Q_{bal}$ did not differ from identity for all measured solutes (Fig 3).

"Estimated" (instant calculations) vs "measured" (full balances) effluent volumes were equal ($BHD_{lf}$: 122.9±0.5 vs 122.7±3.6 l; $BHD_{hf}$: 123.5±0.5 vs 125.5±0.4 l; HDF: 147.3±0.4 vs 148.5±4.7l, p = NS for all) while in NSO measured volume was used for all calculations; since $Q_{inf}$ and $Q_{bal}$ were also equal, differences in Kt (balances vs instant) could be only accounted for by different TAC calculations ($TAC_{bal}$ as compared to $TAC_{inst}$), as shown in Table 3. This difference has two main components: the so-called cardio-pulmonary recirculation occurring in the course of high-efficiency hemodialysis [10], which entails arterial concentration of removed solutes to be lower than in "systemic" blood. Any recirculation at the blood access may also fall into this "disequilibrium". The second component is related to the end-dialysis correction of plasma levels for hemoconcentration and equilibrations for calculation of $TAC_{bal}$. Recalculation of $Kt_{inst}$ by dividing $Q_{inst}$ (according to $D/P_{inst}$ and $TAC_{inst}$) by $TAC_{bal}$, giving a sort of "corrected" $Kt_{inst}$ (indicated as $Kt_{cor}$), gave almost overlapping data as $Kt_{bal}$ (Table 3 and S1 Fig).

Comparison of urea $Kt_{bal}$ and urea Kt derived from Daugirdas' eKt/V ($Kt_{Daug}$) showed equal values in all technologies, with small, non-significant differences in NSO (Table 4), conversely Kt/V calculated by dividing $Kt_{bal}$ by calculated V ($eKt/V_{bal}$) was very close to

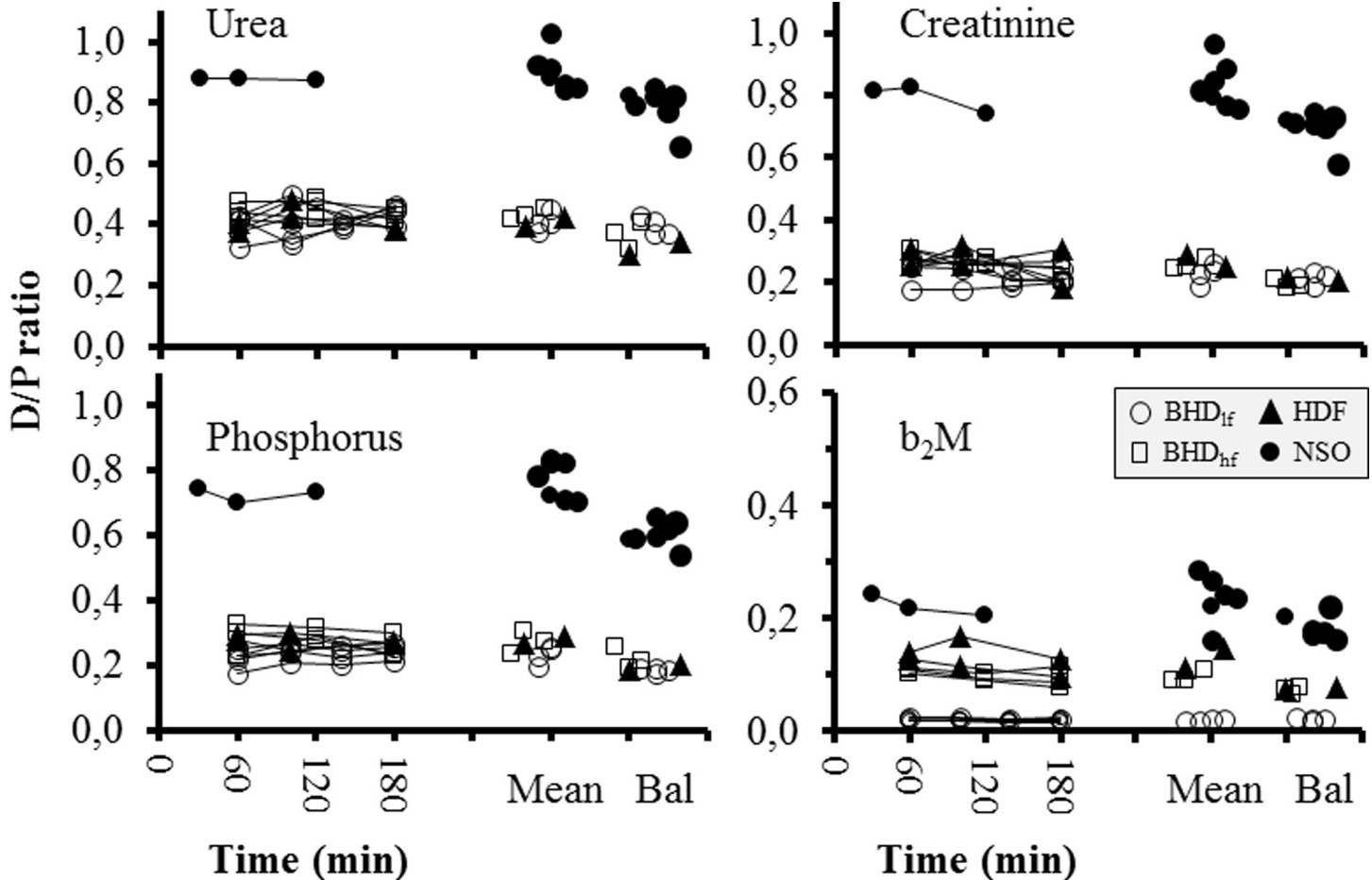

**Fig 1. D/P values "instant" and in full balances.** D/P ratios of urea, creatinine, phosphorus and $b_2M$ are shown as instant data at different treatment times, as mean of all intra-session values (Mean), and as corresponding values in full balance studies (Bal). For NSO, repeated instant calculations are available in a single study, and "Mean" points represents instant calculations at 60 min. Inset indicates treatment modality.

Daugirda's eKt/V, with a correlation equations in all studies near to identity for both Kt (y = 1.04x-2.3, r = 0.99) and eKt/V (y = 0.99x-0.03, r = 0.99) (S2 Fig). Thus our measured urea $Kt_{bal}$, and implicitly V urea are the same as those expressed by the $eKt/V_{Daug}$ equation, bur can be solved separately. In Table 4 weekly urea $Kt_{bal}$ ($wKt_{bal}$, i.e. the sum of 3 weekly treatments in BHD and HDF and 6 treatments in NSO) is also shown: despite not statistically significant differences in stKt/V (Leypold's equation) [11], $wKt_{bal}$ in NSO was significantly lower than in $BHD_{lf}$ and marginally lower than in $BHD_{hf}$ and HDF.

**Table 1. Mean±SD of D/P, Kt and whole session solute removal (Q) according to instant evaluations ("inst"; mean value for each study) and full balance calculations ("bal") in 4 $BHD_{lf}$, 3 $BHD_{hf}$ and 2 HDF.**

| | $D/P_{inst}$ | $D/P_{bal}$ | | $Kt_{inst}$ | $Kt_{bal}$ | | $Q_{inst}$ | $Q_{bal}$ | |
|---|---|---|---|---|---|---|---|---|---|
| | mg/mg | mg/mg | p | l/4hrs | l/4hrs | p | g/session | g/session | p |
| Urea | 0.42±0.02 | 0.37±0.04 | 0.01 | 53.5±5.0 | 46.7±4.3 | 0.01 | 26.3±12.7 | 27.2±9.8 | 0.66 |
| Creatinine | 0.25±0.03 | 0.21±0.02 | 0.01 | 31.8±5.7 | 26.6±3.4 | 0.01 | 1.300±0.522 | 1.355±0.512 | 0.01 |
| Phosphorus | 0.26±0.03 | 0.20±0.02 | 0.01 | 32.9±5.8 | 25.7±3.6 | 0.01 | 0.822±0.205 | 0.831±0.228 | 0.67 |
| $b_2M$ | 0.07±0.05 | 0.05±0.03 | 0.04 | 9.3±7.1 | 6.7±4.0 | 0.05 | 0.143±0.083 | 0.154±0.093 | 0.03 |

**Table 2. Mean±SD of D/P, Kt and solute removal (Q) in 7 NSO balance studies according to instant ("inst") evaluation at 60 min and full balance ("bal") calculations.**

| | $D/P_{inst}$ | $D/P_{bal}$ | | $Kt_{inst}$ | $Kt_{bal}$ | | $Q_{inst}$ | $Q_{bal}$ | |
|---|---|---|---|---|---|---|---|---|---|
| | mg/mg | mg/mg | p | l/session | l/session | p | g/session | g/session | p |
| Urea | 0.90±0.06 | 0.79±0.06 | 0.01 | 22.3±3.8 | 19.4±3.6 | 0.01 | 23.0±7.0 | 20.7±5.0 | 0.13 |
| Creatinine | 0.83±0.07 | 0.70±0.06 | 0.01 | 20.6±3.1 | 17.3±3.0 | 0.01 | 1.363±0.172 | 1.223±0.117 | 0.04 |
| Phosphorus | 0.77±0.06 | 0.61±0.04 | 0.01 | 19.1±2.7 | 15.1±2.9 | 0.01 | 0.859±0.357 | 0.746±0.305 | 0.03 |
| $b_2M$ | 0.24±0.05 | 0.19±0.02 | 0.04 | 6.1±1.6 | 4.7±0.9 | 0.03 | 0.122±0.033 | 0.111±0.031 | 0.60 |

Even though studies in each technology were few, we could show significant differences in removal efficiency (i.e. Kt) of individual solutes between methodologies (S1 Table): $b_2M$ Kt was lower, as expected, in $BHD_{lf}$ than in all other high-flux membrane-based methodologies, creatinine and phosphorus were higher than in $BHD_{lf}$ in HDF and NSO but not in $BHD_{hf}$,

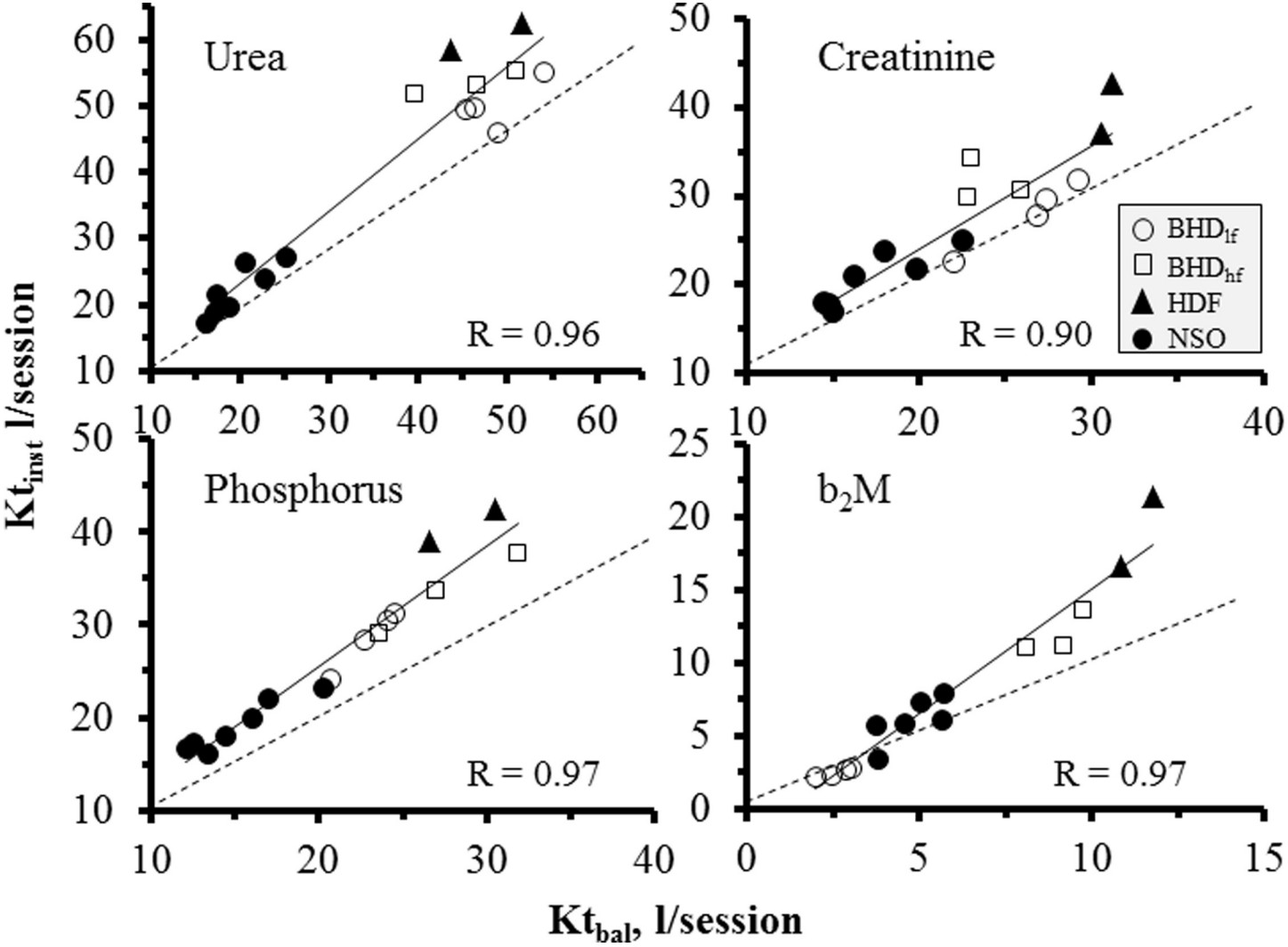

**Fig 2. Correlation between $Kt_{bal}$ and $Kt_{inst}$.** Individual data for urea, creatinine, phosphorus and $b_2M$ is shown. Correlation equation lines (full) and coefficients, and identity lines (interrupted) are indicated. Inset indicates treatment modality.

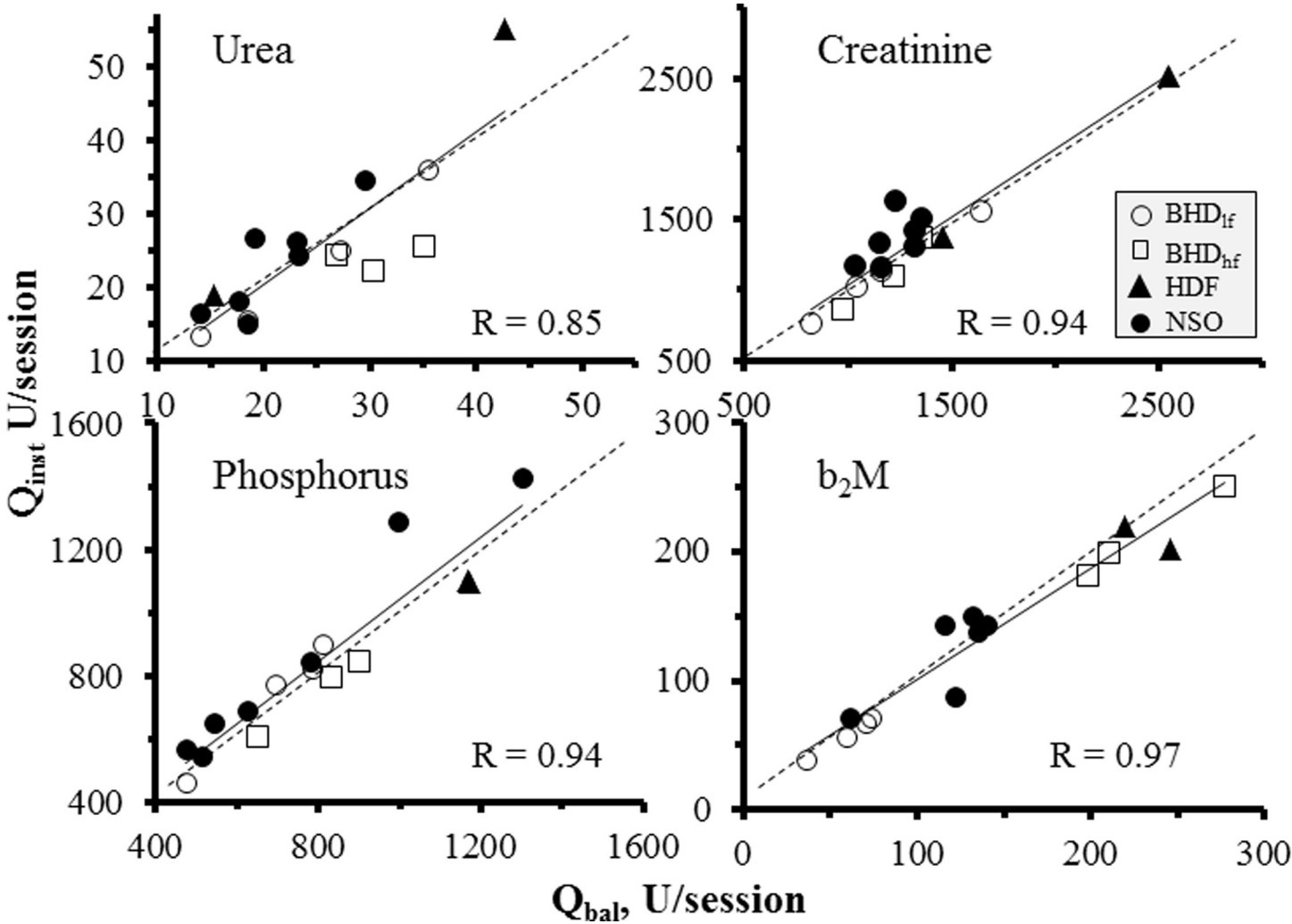

**Fig 3. Correlation between measured ($Q_{bal}$) and calculated ($Q_{inst}$) solute quantity in spent dialysate.** Individual data for urea, creatinine, phosphorus and $b_2M$ is shown; $Q_{bal}$ is measured in full dialysate collection, $Q_{inst}$ is estimated quantity by "instant" calculations. Correlation equation lines (full) and coefficients, and identity lines (interrupted) are indicated. U indicates unit of measure (g for urea, mg for all other). Inset indicates treatment modality.

and urea Kt was lowest in NSO. Instant data gave substantially similar qualitative information as full balance data. Interestingly enough, taking urea Kt as reference within each methodology (i.e. supposing to equalize each technology for urea Kt) a "fractional clearance" or "fractional

**Table 3. Comparison of urea, creatinine, phosphorus and $b_2M$ (mean±SD) time-averaged plasma levels calculated from "instant" samples ($TAC_{inst}$), or level at time 60 (in NSO, $P_{60}$), with that calculated from start and end (with correction for hemoconcentration and equilibration) values in full balances ($TAC_{bal}$), and of Kt calculated from full balances ($Kt_{bal}$) with "corrected" Kt ($Kt_{cor}$) (see text), in BHD/HDF (n = 9) and NSO studies (n = 7).**

| | BHD/HDF | | | | | | NSO | | | | | |
|---|---|---|---|---|---|---|---|---|---|---|---|---|
| | $TAC_{inst}$ | $TAC_{bal}$ | p | $Kt_{cor}$ | $Kt_{bal}$ | p | $P_{60}$ | $TAC_{bal}$ | p | $Kt_{cor}$ | $Kt_{bal}$ | p |
| | mg/dl | mg/dl | | l/run | l/run | | mg/dl | mg/dl | | l/run | l/run | |
| Urea | 48.5±20.8 | 58.9±22.7 | 0.02 | 44.7±9.6 | 46.7±4.3 | 0.5 | 102.6±22.7 | 107.3±18.2 | 0.2 | 21.7±6.3 | 19.4±3.6 | 0.11 |
| Creatinine | 4.01±0.97 | 5.03±1.44 | 0.01 | 25.4±3.7 | 26.6±3.4 | 0.01 | 6.64±0.40 | 7.41±1.14 | 0.05 | 18.9±4.5 | 17.3±3.0 | 0.09 |
| Phosphorus | 2.48±0.42 | 3.29±0.42 | 0.01 | 25.0±3.0 | 25.7±3.6 | 0.43 | 4.38±1.22 | 5.10±1.25 | 0.02 | 16.6±4.0 | 15.1±2.9 | 0.06 |
| $b_2M$ | 1.87±0.58 | 2.34±0.47 | 0.03 | 6.2±3.6 | 6.7±4.0 | 0.08 | 2.06±0.48 | 2.48±0.56 | 0.01 | 5.0±1.4 | 4.7±0.9 | 0.4 |

**Table 4. Comparison of urea removal indices (mean±SD) according to standard plasma kinetic model (Daugirdas [9] and Leypold [11] equations) and our full balance calculations (bal) in different dialytic technologies.** $Kt_{bal}$ is indicated as single session (l/session) and as weekly extrapolation (wKt, l/week).

| | $Kt_{bal}$ | $Kt_{Daug}$ | p | $eKt/V_{bal}$ | $eKt/V_{Daug}$ | p | stKt/V | $_wKt_{bal}$ |
|---|---|---|---|---|---|---|---|---|
| BHDlf (n = 4) | 48.0±4.0 | 48.5±4.4 | 0.9 | 1.52±0.25 | 1,53±0.23 | 0.9 | 2.46±0.13 | 146.1±14.0 |
| BHDhf (n = 3) | 45.7±7.7 | 45.1±5.1 | 0.9 | 1.43±0.14 | 1.41±0.18 | 0.9 | 2.35±0.14 | 137.1±17.0 |
| HDF (n = 2) | 45.6±3.8 | 45.0±4.5 | 0.9 | 1.62±0.04 | 1.48±0.05 | 0.9 | 2.41±0.04 | 136.7±14.4 |
| NSO (n = 7) | 19.4±3.6* | 17.8±1.9* | 0.08 | 0.56±0.18* | 0.51±0.13* | 0.9 | 2.45±0.49 | 116.2±21.6˚ |

*p<0.01 vs $BHD_{lf\&hf}$ and HDF

˚ p<0.04 vs $BHD_{lf}$

removal" of each non-urea solute could be calculated as the ratio of "solute" Kt over "urea" Kt (S1 Table): Fig 4 shows significantly higher creatinine and phosphorus fractional clearances in NSO than in all other methodologies, of phosphorus in HDF as compared to $BHD_{lf}$, and of $β_2M$ in all high-flux membrane-based technologies as compared to $BHD_{lf}$. Balance and instant calculations gave concordant information. Thus urea removal efficiency does not translate into proportional efficiency for non-urea solutes in different methodologies.

## Discussion

Urea Kt, representing blood volume totally cleared of urea by each dialysis treatment, has been proposed as a better measure of dialysis dose than indices factored for V (i.e. putative urea

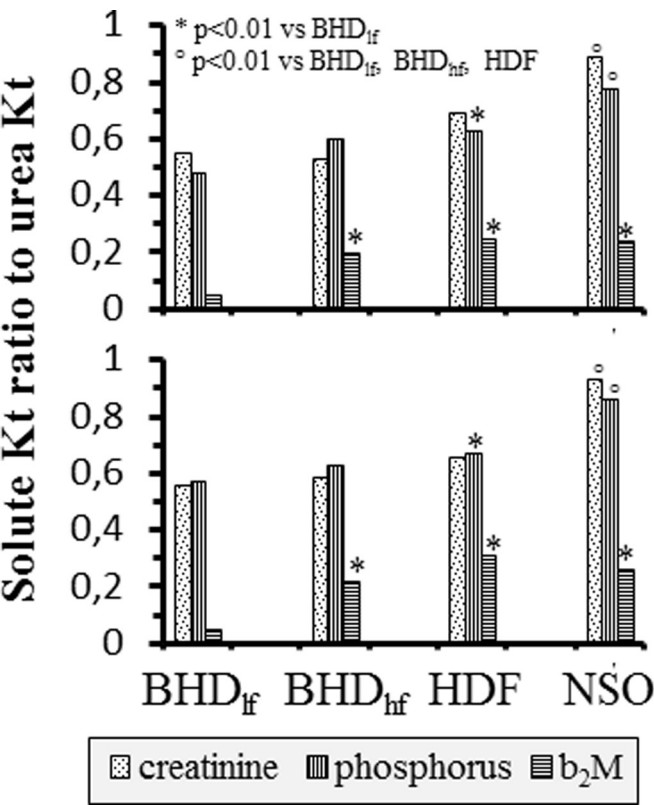

**Fig 4. Ratio of individual non-urea solutes Kt to urea Kt in each dialytic methodology according to full balance studies (top) or instant calculations (bottom).** Individual solutes and significant differences are indicated in the graph.

distribution volume); in effect in large cohorts of dialyzed patients it was shown to predict mortality better than Kt/V, especially in patients at the extreme ranges of body weight [12–15]. Calculation of urea Kt was derived either from Kt/V (calculated according to urea plasma kinetic models) and independent evaluation of V [12, 13], or more recently by on line ionic dialysance-based urea clearance multiplied by time on treatment [14, 15].

We show here that Kt may be measured in the clinical settings for any solute removed by dialysis; use of this factor may help investigation on dialytic efficiency to be expanded beyond that of urea. While we are unable to predict that any dialysis dosing according to non-urea solutes might show better prediction of clinical outcomes than urea itself, we suggest a method for deriving new indices to test for such prediction. In the immediate, we show that Kt allows quantitative measure of dialysis efficiency and a means to compare performance of treatments based on differentiated modalities (length, frequency, convection vs. diffusion, blood to dialysate flow ratio among others) and technologies (membrane characteristics, surface, dialyzer and circuit design, etc.). We show indeed that removal efficiency may be solute and technology-dependent: on the one hand single session Kt of individual solutes showed significant differences between standard prescriptions of different modalities (e.g phosphorus and $b_2M$ Kt was lower in $BHD_{lf}$ than in HDF and NSO), and on the other hand the proportion of individual solute clearances to urea clearance (a so called "fractional clearance") within each technology was not homogeneous across modalities. Thus, urea-based measures of dialytic efficiency (from which adequacy indices are derived) may not translate into proportional removal efficiency for other solutes. This has long been known for large solutes (the so called middle molecular weight solutes, such as $b_2M$), here we show that this may hold true also for small, easily diffusible molecules, such as phosphorus and creatinine, and derive a method for quantifying these differences. This quantitative direct measure of removal efficiency may be applied in the comparison of novel strategies and technological solutions with traditional methodologies, such as "more frequent" treatments against thrice-weekly treatments [2–4] and, more recently, high/medium cut-off membranes against traditional HF membranes [16,17].

To prove validity to our data, we have shown that our measured single treatment urea Kt gives almost an identical quantity as that derived by the familial blood-based eKt/V equation; while the eKt/V equation does not allow to be independently solved for Kt and V, we could independently derive the V factor from measured urea in spent dialysate and corresponding changes in plasma levels and thus solve for eKt. The equivalence of our urea $Kt_{bal}$ with the eKt factor in Daugirdas equation makes it plausible the translation of Kt to any solute as an accurate measure of volume of plasma cleared by that solute by any dialysis modality. Unlikely eKt/V, Kt is an "absolute" quantity, or "absolute dialysis dose", independent from patient demographics, and represents mostly the work performed by technology in use (i.e. its "efficiency"). To translate Kt into an "adequacy" index, or "normalized dialysis dose", it should be factored for any sort of patient categorization in common clinical use, e.g. body water, body weight or body surface area, suggested as the best reference to normalize dose [2, 14, 15]. In the same way Kt is not a quantitative removal measure, which is instead driven by prevalent solute plasma levels [3, 4] and can indeed be calculated with good approximation as the product of Kt by an appropriate TAC determination (Fig 3).

For high dialysate volume-based technologies routine dialysate collection for solute measurement remains unfeasible, which is why historically blood-based models for evaluation of dialytic efficiency were developed and became the standard metrics of dialysis dose. The most relevant component of our study is to show that a single, or better a few instant contemporary collections of spent dialysate and blood may allow a simple and reliable calculation of whole session Kt by multiplying the D/P ratio by total session dialysate volume (a machine-offered information), thus avoiding whole dialysate collection. D/P was shown to remain stable along

treatments; the coefficient of variation between 3–4 intra-run determinations from 60 to 180 min remained within a 16% range in most determinations. Taking the mean of 3–4 intra-run determinations for calculating the Kt, we could verify a nice correlation between $Kt_{inst}$ and measured Kt in full balance determinations (i.e. $Kt_{bal}$), with a systematic trend of the former to overestimate the latter as a consequence of the well-known "disequilibrium" during high-efficiency dialysis between arterial and "systemic" blood solute concentration [10]. In effect, while "instant" blood samples were drawn at the arterial line without any blood flow reduction, $TAC_{bal}$ was calculated from determinations taken pre- and post-dialysis after low-flow, stop-dialysis for 5 min to dissipate any cardio-pulmonary or access-related recirculation [10] and with additional correction for hemoconcentration and "equilibration" with tissue and intracellular spaces. An indirect confirmation of this "disequilibrium" explanation for the differences between $Kt_{inst}$ and $Kt_{bal}$ comes from similar $b_2M$ $Kt_{inst}$ and $Kt_{bal}$ in $BHD_{lf}$ (Fig 2), where membranes in use allow only minimal $b_2M$ removal and consequently entail almost no arterial-venous concentration disequilibrium [10]. Thus, while calculations in full balance studies represent "total body" clearances and effects, instant calculations represent the process of the dialytic system on blood delivered to dialyzer. As a proof of concept, quantitative solute removal derived by $Kt_{inst}$ multiplied by average arterial blood concentrations during dialysis ($TAC_{inst}$) matches measured quantity in the whole collected dialysate. Thus, the "machine" parameter (i.e. $Kt_{inst}$) might also be reverted to a "patient" parameter (i.e. $Kt_{bal}$) by dividing estimated $Q_{inst}$ by a contemporary evaluation of $TAC_{bal}$ (which requires two additional blood samples at start and end-dialysis). In the perspective of Kt measurement as a technology evaluation and comparison, this correction remains fully unnecessary, inasmuch as $Kt_{inst}$ maintains its specific meaning of technological performance.

It has to be anticipated that reliability of our simplified Kt calculations is based on specific operative requirements: the run should be proceeding at a stable cruise rate (concerning mainly dialysate, blood and UF rates) at the time of the sampling and for most time of treatment; significant recirculation at blood access should be excluded by independent methods as part of the routine dialysis supervision in time. Since all our patients had a well-functioning native fistula as vascular access, we are actually unable to extend our observations to patients with central catheters. While our present calculations were all based on plasma water concentrations, we think that avoidance of such correction might also give acceptable results. For low-volume dialysate modalities (either DP or low-dialysate hemodialysis), a whole dialysate collection and sampling remains an easier and better procedure to calculate Q and Kt.

Weekly Kt extrapolation, a simple summation of all individual treatments, may better allow to compare modalities with different frequency/time than single session Kt, constituting an integration over a reasonably homogeneous treatment cycle [2]. Since in dialysis patients body solute pool and plasma levels increase up to and stabilize around a level were removal matches generation, at similar generation rates differences in plasma clearance (i.e. Kt) are expected to result in parallel differences in plasma levels. We show indeed (Tables 3 and 4) that despite similar stKt/V, urea wKt was lower in NSO than in BHD and HDF, and this was attended by higher $TAC_{urea}$. Weekly urea removal was not different between technologies (Tables 1 and 2), confirming that differences in Kt predict parallel differences in plasma levels.

Kt based on instant D/P measure has limitations: the most important one is that while whole dialysate collection is a reliable summary of a whole run, instant D/P may miss periods of unequal efficiency along the run due to any occasional changes in dialysis settings. It is the operator's responsibility to judge on the uniformity/non-uniformity of the run. An average of several instant D/P along the run are more convenient than a single determination to limit variability. Additionally, dialysate-side calculations (both "inst" and "bal") are expected to underestimate total removal of peptides/small proteins being absorbed to dialyzer membranes, however small

this phenomenon may be [4]. Lastly, we have not checked for D/P stability along the run and detectability in dialysate of poorly diffusible, tightly protein-bound solutes, blamed of significant uremic toxicity. Due to their putative relevance, further investigation on extendibility of our methodology to these compounds appears worthwhile. The strength of our data is that it allows in the real world practice a simple and reliable quantitative measure of the efficiency with which any free plasma solute, as long as it is detectable in the dialysate, is removed by dialysis itself without the need of whole dialysate collection. Target single session urea Kt values have already been suggested in tabular form as possible indices of dialysis adequacy [14, 15], but for other solutes those limits have to be defined. Irrespective of any adequacy meaning, Kt metrics represents a novel, powerful instrument for quantitative comparisons between methodologies, and may be applied to check at a patient level, rather than in "ex-vivo" simulations, performance of the many technologies and dialysis methodologies already in clinical use or forthcoming.

## Conclusions

We present here a simple method to calculate a Kt factor representing the volume of blood being totally cleared of individual solutes by any dialytic modality, and validate our simplified calculations by comparison with calculations in full dialysate collection. Extrapolation of single session Kt to weekly calculations allows objective comparisons to be made between technologies/modalities based on different frequency/time.

## Supporting information

**S1 File. Appendix.** Calculation of urea distribution from quantity in spent dialysate and changes in plasma levels.
(DOCX)

**S2 File. Correlation analysis full results.**
(DOCX)

**S1 Fig. Correlation between Kt$_{bal}$ and Kt$_{cor}$.** Kt$_{cor}$ is calculated from Q$_{inst}$ and TAC$_{bal}$, as explained in the text. Individual data is shown for urea, creatinine, phosphorus and b$_2$M. Correlation equation lines (full) and coefficients, and identity lines (interrupted) are indicated. Treatment modality is indicated in the inset.
(TIF)

**S2 Fig. Correlation between urea eKt/V (top) and Kt (bottom) according to Daugirdas' equation and our calculation in full "balance" studies.** eKt/VDaug and Kt$_{Daug}$ indicate values derived by Daugirdas' second generation equation, eKt/Vbal and Kt$_{bal}$ refer to our calculations in full balance studies. Correlation equation lines (full) and coefficients, and identity lines (interrupted) are shown. Treatment modality is indicated in the inset.
(TIF)

**S1 Table. Weekly Kt (mean±SD, liters) of urea, creatinine, phosphorus and β$_2$M with different dialytic modalities and calculations ("bal" and "inst").** In parenthesis is the ratio of mean Kt in different modalities against corresponding BHD$_{lf}$ mean, taken as reference. Ratio to urea expresses the ratio of individual non-urea solute Kt against Kt of urea within each dialytic modality for each study (mean±SD). Statistical level of differences is indicated below the table.
(DOCX)

**S1 Dataset.**
(XLS)

## Acknowledgments

We wish to thank the nurses Roberta Pivetta and Franca Cassaro for skillful attendance to all balance studies and passionate patient training to home hemodialysis, the nurse Giuliana Carta for collaborating in balance performance and the staff coordinators Paola Cortesi and Filomena Marino for careful organization and supervision.

## Author Contributions

**Conceptualization:** Giacomo Colussi, Chiara Carla Maria Brunati.

**Data curation:** Chiara Carla Maria Brunati, Francesca Gervasi, Alberto Montoli, Denise Vergani, Federica Curci.

**Formal analysis:** Giacomo Colussi.

**Investigation:** Chiara Carla Maria Brunati, Alberto Montoli.

**Methodology:** Giacomo Colussi, Chiara Carla Maria Brunati.

**Project administration:** Giacomo Colussi, Chiara Carla Maria Brunati.

**Resources:** Enrico Minetti.

**Supervision:** Chiara Carla Maria Brunati, Francesca Gervasi, Alberto Montoli, Denise Vergani, Federica Curci.

**Validation:** Alberto Montoli.

**Visualization:** Francesca Gervasi, Denise Vergani.

**Writing – original draft:** Giacomo Colussi.

**Writing – review & editing:** Giacomo Colussi, Chiara Carla Maria Brunati, Alberto Montoli, Enrico Minetti.

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
