## [Decision Letter · Decision Letter 0]

15 Apr 2020

PONE-D-20-05583

A simple method to measure the Kt factor for comparing removal efficiency of uremic retention solutes with different dialytic technologies

PLOS ONE

Dear Dr Colussi,

Thank you for submitting your manuscript to PLOS ONE. After careful consideration, we feel that it has merit but does not fully meet PLOS ONE’s publication criteria as it currently stands. Therefore, we invite you to submit a revised version of the manuscript that addresses the points raised during the review process.

Two reviewers have evaluated your manuscript. Although both find it of some interest, several important comments were made that must ALL be adressed in your rebuttal and the revised manuscript. 

To enhance the reproducibility of your results, we recommend that if applicable you deposit your laboratory protocols in protocols.io, where a protocol can be assigned its own identifier (DOI) such that it can be cited independently in the future. For instructions see: http://journals.plos.org/plosone/s/submission-guidelines#loc-laboratory-protocols

We look forward to receiving your revised manuscript.

Kind regards,

Jaap A. Joles, DVM, PhD

Academic Editor

PLOS ONE

Journal Requirements:

2. In the ethics statement in the manuscript and in the online submission form, please provide additional information about the patient records used in your retrospective study, including: a) whether all data were fully anonymized before you accessed them; b) the date range (month and year) during which patients' medical records were accessed; c) the date range (month and year) during which patients whose medical records were selected for this study sought treatment; and d) the source of the medical records analyzed in this work (e.g. hospital, institution or medical center name). If patients provided informed written consent to have data from their medical records used in research, please include this information.

3. Please provide a sample size and power calculation in the Methods, or discuss the reasons for not performing one before study initiation.

4. Thank you for stating the following in the Competing Interests section: "I have read the journal's policy and the authors of this manuscript have the following competing interests: Drs. GC and CCMB have been past medical advisors for NxStage Medical Inc, Lawrence, MA; Drs. CCMB and EEM received fees as speaker in dialysis industry-sponsored symposia at the National Congress of Italian Society of Nephrology. Drs. AM, FG, DV and FC have no conflict of  interests to declare"

Reviewers' comments:

Reviewer's Responses to Questions

**Comments to the Author**

1. Is the manuscript technically sound, and do the data support the conclusions?

Reviewer #1: Yes

Reviewer #2: Partly

2. Has the statistical analysis been performed appropriately and rigorously? 

Reviewer #1: Yes

Reviewer #2: I Don't Know

3. Have the authors made all data underlying the findings in their manuscript fully available?

Reviewer #1: Yes

Reviewer #2: Yes

4. Is the manuscript presented in an intelligible fashion and written in standard English?

Reviewer #1: Yes

Reviewer #2: Yes

5. Review Comments to the Author

Reviewer #1: This paper provides a simple method for quantification of non-urea uremic retention solutes to assess dialysis adequacy. This is very relevant considering the fact that increasing evidence suggests that middle molecules and PBUTs are associated with adverse patient outcome, and should therefore be taken into account when assessing dialysis efficiency. I have the following recommendations for improvement:

1. In line 55-58, calculation of the Kt factor is described. However, when you divide the total quantity removed by time-averaged plasma concentration (TAC) you obtain a plasma clearance in mL/min if I’m correct, whereas previously it was stated that the Kt factor is the plasma volume totally cleared of any solute. Please clarify.

2. Instant D/P and Kt were significantly higher compared with the balanced D/P and Kt. As balanced values are more accurate, and instant and balanced measurement were correlated, would it be possible to apply a mathematical correction to approximate balanced metrics more accurately using only instant measurements? This would reduce blood and dialysate sampling frequency, improving the clinical applicability of this method.

3. To facilitate the interpretation of figures, I would recommend to include a legend on the right side of the figure explaining the meaning of bars and symbols, replacing or complementing the explanation in the legend.

4. If Kt metrics would be used in the clinics to assess dialysis adequacy, targets must be available to guide clinicians on how to adjust the dialysis prescription. Therefore, standardization of Kt for distribution volume/bodyweight/BSA is required. Therefore, the method for calculation of Kt presented in this study is not usable for evaluating dialysis adequacy in patients yet. Although the authors state that Kt can be used for comparison of dialysis technologies/modalities, this should be pointed out more clearly in the discussion.

5. Discussion: it is repeatedly stated that Kt may be measured in the clinical settings for “any” solute removed by dialysis. However, only small and middle molecular weight uremic toxins were evaluated in the present study, and protein-bound uremic toxins (PBUTs) were not. In contrast to small and middle molecular weight uremic toxins, D/P (using the free PBUT fraction for plasma concentration) may not remain stable for PBUTs as the equilibrium between the free and bound PBUT fraction shifts towards the bound-fraction during a dialysis session. If possible, you could consider measuring PBUTs if storage samples are available. Otherwise, please limit the statement to small and middle molecular weight toxins.

6. English editing (language and grammar) is required.

Reviewer #2: (1) The paper compares NSO treatments (with high flux filters) versus BHD/HDF treatments (both with low flux and high flux, with and without ultrafiltration). The title of the paper could also have been "differences between NSO and in-center HD modalities". Apparently the authors selected the frequency of the treatment as main difference, but it can be argued that the main differences in treatment are the use of low flux versus high flux filters, or the amount of ultrafiltration. Next to (or instead) a comparison between NSO and BHD/HDF a comparison between low flux and high flux filters could (should) have been presented and discussed in more detail (for all the measured toxins urea/creat/phosphate/b2m).

(2) It is not entirely clear how the "inst" values (Ktinst, Qinst) and "bal" values (Ktbal, Qbal) have been derived. A better explanation than the given "intra-session mean of instant points" and "calculated values form full balance data" well help the reader to understand what these values really mean.

(3) It is recognized that "Kt" is better measure to compare the effectiveness of a specific dialysis treatment (such as BHD, HDF, NSO, PD) in stead of "Kt/V". But as a measure of dialysis dose to a specific patient "Kt/V" has to be preferred. Perhaps it is better to talk about Kt as "absolute dialysis dose" and Kt/V as "normalized dialysis dose".

4) The discussion paragraph is quite long and extensive. For clarity and readiness it would be better to condense and to focus on the main points.

6. PLOS authors have the option to publish the peer review history of their article (what does this mean?). If published, this will include your full peer review and any attached files.

Reviewer #1: No

Reviewer #2: No

---

## [Author Response · Author response to Decision Letter 0]

20 Apr 2020

Please refer to the "Response to Reviewrs" letter

---

## [Decision Letter · Decision Letter 1]

5 May 2020

A simple method for the calculation of dialysis Kt factor as a quantitative measure of removal efficiency of uremic retention solutes:  applicability to high-dialysate vs low-dialysate volume technologies

PONE-D-20-05583R1

Dear Dr. Colussi,

We are pleased to inform you that your manuscript has been judged scientifically suitable for publication and will be formally accepted for publication once it complies with all outstanding technical requirements.

With kind regards,

Jaap A. Joles, DVM, PhD

Academic Editor

PLOS ONE

Additional Editor Comments (optional):

Reviewers' comments:

Reviewer's Responses to Questions

**Comments to the Author**

1. If the authors have adequately addressed your comments raised in a previous round of review and you feel that this manuscript is now acceptable for publication, you may indicate that here to bypass the “Comments to the Author” section, enter your conflict of interest statement in the “Confidential to Editor” section, and submit your "Accept" recommendation.

Reviewer #1: All comments have been addressed

Reviewer #2: All comments have been addressed

2. Is the manuscript technically sound, and do the data support the conclusions?

Reviewer #1: Yes

Reviewer #2: Yes

3. Has the statistical analysis been performed appropriately and rigorously? 

Reviewer #1: Yes

Reviewer #2: Yes

4. Have the authors made all data underlying the findings in their manuscript fully available?

Reviewer #1: Yes

Reviewer #2: Yes

5. Is the manuscript presented in an intelligible fashion and written in standard English?

Reviewer #1: Yes

Reviewer #2: Yes

6. Review Comments to the Author

Reviewer #1: The authors have addressed all of my comments and I now find the manuscript acceptable for publication.

Reviewer #2: (No Response)

7. PLOS authors have the option to publish the peer review history of their article (what does this mean?). If published, this will include your full peer review and any attached files.

Reviewer #1: Yes: Maaike K. van Gelder

Reviewer #2: No

---

## [Editor Report · Acceptance letter]

20 May 2020

PONE-D-20-05583R1 

A simple method for the calculation of dialysis Kt factor as a quantitative measure of removal efficiency of uremic retention solutes: applicability to high-dialysate vs low-dialysate volume technologies 

Dear Dr. Colussi:

I am pleased to inform you that your manuscript has been deemed suitable for publication in PLOS ONE. Congratulations! Your manuscript is now with our production department. 

With kind regards,

on behalf of

Dr. Jaap A. Joles 

Academic Editor

PLOS ONE